# Electronic Effects in Cobalt Phthalocyanine Catalysts Towards Noble-Metal-Free, Photocatalytic CO_2_-to-CO Reduction

**DOI:** 10.3390/molecules29214994

**Published:** 2024-10-22

**Authors:** Fan Ma, Hong-Wei Lin, Zizi Li, Wen-Jing Li, Jia-Wei Wang, Gangfeng Ouyang

**Affiliations:** 1School of Chemical Engineering and Technology, Sun Yat-sen University, Zhuhai 519082, China; mafan3@mail2.sysu.edu.cn (F.M.); linhw29@mail2.sysu.edu.cn (H.-W.L.); lizz7103@163.com (Z.L.); liwj358@mail2.sysu.edu.cn (W.-J.L.); cesoygf@mail.sysu.edu.cn (G.O.); 2Chemistry College, Center of Advanced Analysis and Gene Sequencing, Zhengzhou University, Zhengzhou 450001, China; 3Guangdong Provincial Key Laboratory of Emergency Test for Dangerous Chemicals, Guangdong Institute of Analysis (China National Analytical Center Guangzhou), Guangzhou 510070, China

**Keywords:** cobalt phthalocyanine, CO_2_ reduction, electronic effect, copper photosensitizer, noble-metal-free system

## Abstract

Noble-metal-free CO_2_ reduction systems based on cobalt phthalocyanine (**CoPc**) and its derivatives have demonstrated remarkable photocatalytic performances; however, their structure-activity relationship with electronic tuning remains unexplored. Herein, we now provide a systematic study to investigate the electron effects of substituents on the **CoPc** family in photocatalytic CO_2_ reduction, where a Cu(I) heteroleptic photosensitizer is utilized. The highest performance can be achieved using cobalt tetracarboxylphthalocyanine in light-driven CO_2_-to-CO reduction, with a maximum turnover number of 2950 at 450 nm and an outstanding apparent quantum yield of 63.5% at 425 nm, over ten times the activity with the tetra-dimethylamino-substituted **CoPc** derivative. The favorable electron-withdrawing effects have been further verified by DFT calculations and cyclic voltammetry, which reduces the overpotential required for CO_2_ reduction and decreases the Gibbs free energy of the catalyst active intermediates, particularly the CO-desorption energetics.

## 1. Introduction

As carbon neutrality and sustainable development are increasingly emphasized, the conversion of inert carbon dioxide (CO_2_) into carbon monoxide (CO) or other recycled substances can be regarded as a way to slow down global warming [1,2,3]. It has been verified that a homogeneous catalytic system based on molecular catalysts can achieve efficient and selective CO_2_-to-CO reduction under illumination. In particular, precious metal complex catalysts have excellent conversion efficiency, such as with Re and Ru [4,5,6]. Yamazaki and Ishitani synthesized Os(II)-Re(I)-Ru(II) tri-core novel supramolecular photocatalyst using a progressive Mizoroki-Heck reaction, which induced CO_2_ reduction under 730 nm visible light with high selectivity and durability. And the turnover number (TON) of CO formation exceeds 4300 [7]. However, if precious metals cannot be recycled efficiently, they are not only rare and expensive, destined to fail to achieve large-scale applications, but also violate the original intention of protecting the environment and promoting sustainable development. To date, more attention has been paid to the construction of noble-metal-free systems, mostly based on the earth-abundant metals, including Cu [8], Ni [9], Co [10], and Mn [11,12]. The TON of non-precious metals in N, N-dimethylformamide with [Fe(qpy)(OH_2_)_2_]^2+^ (qpy = 2,2′:6′,2″:6″,2‴-quaterpyridine) as a catalyst and purpurin as the photosensitizer (PS) reached 1365, with an apparent quantum yield (*Φ*) of 1.1% [13]. When iron porphyrin was paired with aminoanthraquinone, the TON could be up to 16,100 and *Φ* could be up to 11.1%, driven by light [14]. However, these examples suggest that there is still room for improvement in the *Φ* of earth-abundant photocatalytic systems.

**CoPc** derivatives constitute an attractive class of electrocatalysts for CO_2_ reduction, with distinct advantages in terms of accessibility, stability, and structural tunability. Notably, though **CoPc** performs well in the electrocatalytic CO_2_ reduction [15,16,17], it has been less used in light-driven systems. For the first time, we achieved an *Φ* of 10.2% with a TON of 397 by combining an Ir(III) PS with **CoPc** for selective CO_2_-to-CO conversion [18]. The *Φ* could be further optimized to 27.9% by decorating amino groups as the beta-substituents on **CoPc**. To develop earth-abundant photocatalytic systems, we recently attempted to develop photochemical systems using the **CoPc** derivatives while minimizing the use of precious metals. In these studies, [Cu^I^(xantphos)(bcp)]PF_6_ (**CuBCP**; xantphos = 9,9-dimethyl-4, 5-bis(diphenylphosphino)xanthene, bcp = bathocuproine) was adopted as the replacement of the noble-metal PS [19,20]. Our initial approach was to improve the performance of photocatalytic CO_2_-to-CO conversion by the installation of fluorine substituents on **CoPc**, which increased the *Φ* from 19.0% to 58.2%, along with giving a maximum TON of 9185 [19]. The electron-withdrawing effect of the fluorine substituents seemingly reduced the overpotential required for driving the reaction. Subsequently, **CoTCPc** was used as another promising molecular catalyst, resulting in an outstanding *Φ* of 76%, with a maximum turnover number of 11,800, mainly boosted by the dynamic coordinative interactions with a pyridine-decorated Cu photosensitizer [20]. Nonetheless, the high activity could also be based on the electron-withdrawing carboxyl groups of cobalt tetracarboxylphthalocyanine (**CoTCPc**), as indicated by the electrochemical and DFT calculations of its proposed mechanism. The above preliminary investigations into the electronic effects of **CoPc**-based catalysts in photocatalytic CO_2_ reduction have been presented as isolated cases, while a systematic study on the structure–activity relationship based on a sequence of the **CoPc** catalyst family is desirable for rational guidance for designing molecular catalysts [21,22].

In this context, we present five **CoPc**-based catalysts, modified by various electron-donating/withdrawing *β*-substituents, in noble-metal-free systems for CO_2_ photoreduction driven by **CuBCP** as the PS. The activity of these **CoPc**-based catalysts is disclosed in this order: -N(CH_3_)_2_ < -NH_2_ < -H < -HSO_3_ < -COOH, with the highest (**CoTCPc**) achieving a maximum TON of 2950 at 450 nm and an outstanding *Φ* of 63.5% at 425 nm. DFT calculations and cyclic voltametric results have elucidated the electron effects from the different *β*-substituents, which not only affects the overpotential required for driving the catalytic reaction, but also affects the adsorption energetics of * CO and * COOH intermediates at the reduced Co center. The trade-off within the electronic modulations leads to a structure–activity volcano plot of activity vs. electron-withdrawing effects, accounting for the best performances from **CoTCPc**.

## 2. Results and Discussion

### 2.1. Photocatalytic CO_2_ Reduction

First, a systematic series of different **CoPc**-based catalysts (see Figure 1a) with different substituents for electronic modulation were selected or synthesized, namely cobalt *β*-tetrasulphophthalocyanine (**CoTSPc**), **CoTCPc**, **CoPc**, cobalt *β*-tetraaminophthalocyanine (**CoTAPc**), and cobalt *β*-tetra(dimethylamino)phthalocyanine (**CoTDMAPc**). The catalysts which are not commercially available, including **CoTSPc**, **CoTCPc,** and **CoTDMAPc**, were characterized by high-resolution mass spectrometry (HR-MS; Appendix A), elemental analysis (Appendix A) and UV–vis spectroscopy (Appendix A).

After characterization, visible-light-driven CO_2_ reduction with **CuBCP** PS and Co(II) catalysts was carried out to evaluate the catalytic performance of these Co(II) catalysts in a mixture CH_3_CN/TEA and with BIH (1,3-dimethyl-2-phenyl-2,3-dihydro-1H-benzo[*d*]imidazole) as the sacrificial electron donor. Triethylamine (TEA) fundamentally functions as the proton acceptor to deprotonate the oxidized BIH [18]. As photo-induced dissociation of the diphosphine ligand is a commonly observed drawback of heteroleptic [Cu(P^P)(N^N)]^+^ complexes [23,24], a general protocol was followed [25], where 1.0 mM xantphos ligand (2.0 eq. of PS) was added to the catalytic mixture to inhibit its dissociation during photocatalysis. All five Co(II) catalysts enabled the selective formation of CO with only trace amounts of H_2_ and no liquid products, providing a CO selectivity of over 97% (Figure 1b). It should be noted that all components are necessary to afford a substantial amount of CO in the representative **CuBCP**/**CoTCPc** system (Appendix A), and some factors were further considered for a systematic optimization of the photocatalytic system applying the best catalyst, i.e., **CoTCPc**. As depicted by Figure 1c, a decreasing concentration of **CoTCPc** can significantly increase the TON, with a maximum of 2950 at 0.2 μM within 1 h (Table 1, entry 6). Conversely, the CO yield (ca. 110 μmol when deactivated; Appendix A) plateaued when the catalytic concentration exceeded 100 μM [**CoTCPc]**. This is most likely because the increasing **CoTCPc** concentration leads to a stronger absorption of irradiated light at 450 nm and thus precludes the excitation of PS, as indicated by their partially overlapped UV–vis spectra (Appendix A). In addition to catalyst concentration, the addition of a proton source, like phenol, H_2_O, or 2,2′,2″-trifluoroethanol, can significantly promote catalytic activity (Appendix A). This can be explained by the facilitation of proton-dependent chemical steps in CO_2_ reduction [26]. The reaction with a ^13^CO_2_ atmosphere exhibited significant formation of ^13^CO (*m*/*z* = 29; Appendix A), showing that the CO was produced from the reduction of CO_2_ rather than from the decomposition of organic components.

Based on the optimal conditions, **CoTCPc** showed the highest catalytic performance in terms of CO yield, followed by **CoTSPc**, **CoPc**, **CoTAPc,** and **CoTDMAPc** (Figure 1b; Table 1, entries 1–5) under 450 nm irradiation. As a result, an impressive increase of more than ten-fold in catalytic efficiency is achieved from **CoTDMAPc** to **CoTCPc**. Hence, the usage of electron-withdrawing *β*-substituents tends to improve the catalytic performance of Co(II) catalysts, with **CoTCPc** the optimal one, while the more electron-withdrawing **CoTSPc** shows a lower activity. More impressively, by varying the irradiation wavelengths (i.e., 405, 425, and 450 nm) and impairing the intensity to 40 mW cm^−2^, a maximum *Φ* of 63.5% can be obtained at 425 nm (Figure 1d,e). This *Φ* is comparable to state-of-the-art molecular systems for photocatalytic CO_2_ reduction and even better than some noble-metal-based ones like *fac*-[Re{4,4′-(MeO)_2_bpy}(CO)_3_{P(OEt)_3_}]/*fac*-[Re(bpy)(CO)_3_(CH_3_CN)]^+^ [27] (*Φ*_CO_ = 59% at 365 nm), a Ru-Re dyad (*Φ*_CO_ = 45% at 480 nm) [28]. Most importantly, this *Φ* value is the highest value among the noble-metal-free systems, including Cu-purpurin/FeTDHPP [12] (*Φ*_CO_ = 6% at 450 nm; FeTDHPP = chloroiron(III) 5,10,15,20-tetrakis (2′,6′-dihydroxyphenyl)-porphyrin), *p*-terphenyl/Co-cyclam [29] (*Φ*_HCOOH+CO_ = 25% at 313 nm; cyclam = 1,4,8,11-tetraazacyclotetradecane), and [Cu_2_(P_2_bph)_2_]^2+^/*fac*-Mn(4,4′-(OMe)_2_bpy)(CO)_3_Br [11] (*Φ*_HCOOH+CO_ = 57% at 436 nm; P_2_bph = 4,7-diphenyl-2,9-di(diphenylphosphinotetramethylene)-1,10-phenanthroline). The CO production of the **CuBCP**/**CoTCPc** system ceased after 2 h of reaction, most presumably due to the decomposition of the **CuBCP** [23,30], but the high *Φ* of this photocatalytic system is still impressive. Overall, the above photocatalytic results demonstrate the extraordinary performance of the **CuBCP**/**CoTCPc** system for photocatalytic CO_2_-to-CO conversion.

**Figure 1 molecules-29-04994-f001:**
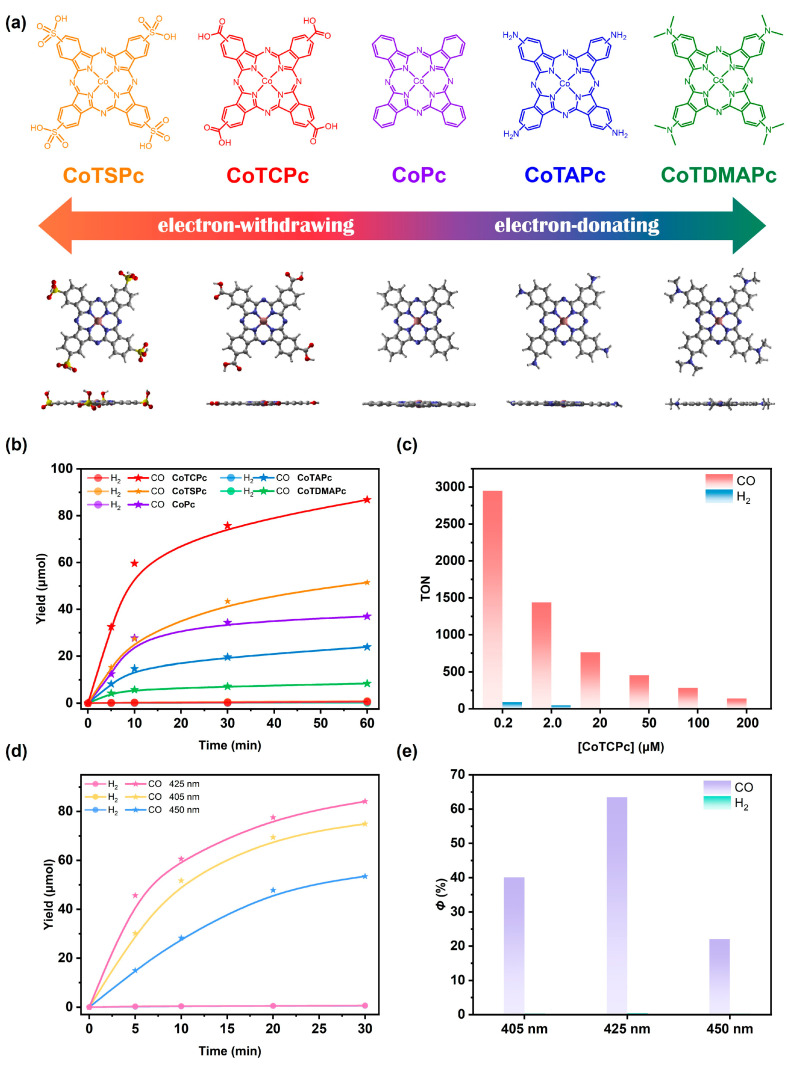
**CoPc**-derived catalysts for photocatalytic CO_2_ reduction. (**a**, top) Chemical and (**a**, bottom) calculated structures (PBE0 [31] functional and def2-SVP basis set; with top and side views) of applied Co(II) catalysts, highlighting electronic effects caused by different *β*-substituents. White, gray, blue, red, and pink colors denote H, C, N, O, S, and Co atoms, respectively. (**b**) Time profiles of photocatalytic CO (star) and H_2_ (circle) formation using 0.5 mM **CuBCP** and 50 μM **CoTCPc** (red), **CoTSPc** (orange), **CoPc** (violet), **CoTAPc** (blue), or **CoTDMAPc** (green) under 450 nm irradiation (100 mW cm^−2^). (**c**) TON comparison for CO and H_2_ formation at varying concentrations of **CoTCPc** with **CuBCP**. (**d**) Wavelength dependency of time profiles of photocatalytic CO (star) and H_2_ (circle) formation with 0.5 mM **CuBCP** and 50 μM **CoTCPc** at 450 (blue), 425 (red), or 405 (green) nm light irradiation (40 mW cm^−2^). (**e**) *Φ* values for CO and H_2_ formation at 5 min of 450, 425, or 405 nm light irradiation (40 mW cm^−2^) from **CuBCP**/**CoTCPc** system.

### 2.2. Electronic Effects of Co(II) Catalysts

As the structures of these *β*-substituted **CoPc** derivatives cannot be defined by single-crystal X-ray diffraction due to the different regioisomers arising from the synthesis with two random *β*-positions, DFT calculations were applied to analyze the structural and electronic effects of the different substituents by fixing them at the positions of 1, 5, 9, and 13. As can be seen from Figure 1a and Table 2, the *β*-substituents result in negligible distortions on the phthalocyanine rings with the nearly 0° angle of plane distortion φ. According to the Hirshfeld population [32] listed in Table 2, the Co atomic charges in these molecules vary through the incorporation of different substituents. All Co atomic charges are positive (Appendix A), which indicates that the electron density is transferred from cobalt to the phthalocyanine substituents. It is found that the Co atomic charges in the complexes with the -N(CH_3_)_2_, -NH_2_, -H, -COOH, and -HSO_3_ substituents are 0.1959, 0.1971, 0.2038, 0.2084, and 0.2122 *e*, respectively. This is in the exact order of the increasing ability of the *β*-substituents to attract electrons (-N(CH_3_)_2_ < -NH_2_ < -H < -COOH < -HSO_3_) and demonstrates the facile modulations of the electronic properties of these Co(II) catalysts. The calculations on the values of HOMO (highest occupied molecular orbital) and LUMO (lowest unoccupied molecular orbital) indicate the small decrease (<0.2 eV) in the HOMO-LUMO gap originating from either electron-donating or electron-withdrawing substitutions on the pristine **CoPc** (Appendix A), while generally decreasing HOMO values could be observed along with the increasing electron-withdrawing effects from the substituents, consistent with the effective electronic modulations on the **CoPc**-derived catalyst family.

Then, cyclic voltametric (CV) methods were utilized to examine their redox properties with respect to the different electron-withdrawing/donating substituents. NMP/TEA (NMP = *N*-methylpyrrolidone; *v*:*v* = 5:1) mixed solvent was chosen as the electrolyte solvent to fully dissolve the **CoPc** derivatives and mimic the alkaline conditions in photocatalysis. Studies of their CVs under N_2_ revealed their first reduction waves at −0.83 to −1.46 V (vs. Fc^+^/Fc^0^ unless otherwise stated), mostly attributable to Co^II/I^ reduction events [18,33]. It has been found that the substituents with stronger electron-withdrawing ability enable the positive shifts of the first reduction wave (Appendix A), further manifesting the electronic effect. Meanwhile, the CVs of these complexes show enhanced current in the presence of CO_2_ and the addition of phenol as an additional proton source, displaying their catalytic capabilities under corresponding conditions in photocatalysis.

The aforementioned electrochemical results have illustrated that the electron-withdrawing effects effectively impose positive shifts of the Co^II/I^ redox couple, which is amenable for photocatalysis at lower overpotentials [19]. However, **CoTSPc**, with the most electron-withdrawing sulfonic-acid substituents, gave a smaller performance than **CoTCPc**. Based on our research into the photocatalytic mechanism of the **CoPc** [19,20], the photocatalytic performance is also related to the binding energies of the active intermediates in **CoPc** catalysts that adsorb * CO and * COOH. To preliminarily decipher the structure–activity relationship, additional DFT calculations (PBE0 [31] functional and def2-SVP basis set) based on the proposed mechanism of **CoPc** were undertaken. The calculated reactions involve the rate-determining generation of * COOH intermediates from the adsorption and reduction of CO_2_ at the Co site. The * COOH intermediate then undergoes a proton-coupled electron transfer step to cleave the C-OH bond and evolves * CO, which is ultimately desorbed to recover the free Co center. We have obtained the calculated structures derived from the used Co(II) catalysts that adsorb CO and COOH with corresponding Gibbs free energies, most importantly the Δ*G*(* COOH) and Δ*G*(* CO) (Appendix A). It can be seen that the values of Δ*G*(* COOH) for all **CoPc** molecules are close, showing a negligible correlation with the order of catalytic activity. In contrast, the order of the Δ*G*(* CO) values is more consistent with the order of catalytic activity (Figure 2), which suggests that the CO desorption step plays a crucial role for the catalytic performance. The key contribution of the CO desorption ability in CO_2_ reduction has been extensively documented [34,35]. This DFT-calculated volcano-type plot computationally indicates **CoTCPc** as the best catalyst in our case, while this primary observation demands more **CoPc** derivatives as well as more detailed theoretical/experimental works, which are in progress, for further confirmation.

### 2.3. Photoinducd Electron Transfer

The overall photocatalytic mechanism of **CoTCPc/CuBCP** has been partially revealed by our previous study [20], where a reductive quenching pathway was proposed. Here, we further confirmed this electron transfer pathway by quenching experiments with the excited state of **CuBCP**, which were carried out by steady-state fluorescent spectroscopy (Figure 3). **CuBCP** can be effectively quenched by BIH (Figure 3a), which suggests a reductive quenching pathway, while the reductive quenching constant for BIH (*k*_q(BIH)_), as determined by Stern–Volmer (S-V) plots, was 5.23 × 10^9^ M^−1^ s^−1^ (Figure 3b). On the other hand, we also observed significant quenching by the addition of **CoTCPc** in the steady-state fluorescent spectra (Figure 3c,d). However, the slopes in S-V plots with **CoTCPc** for **CuBCP** were not linear, with shifted emission peaks. These observations are consistent with the absorption of excitation light and excited luminescence rather than oxidative quenching by **CoTCPc**.

## 3. Materials and Methods

### 3.1. Materials

**CuBCP** [36], BIH [37], **CoTSPc** (deprotonated and protonated) [38], and **CoTCPc** [39,40] were prepared following the previously reported methods. Milli-Q ultrapure water (>18 MΩ) was utilized, unless otherwise stated. [Cu(CH_3_CN)_4_]PF_6_ (97%, Innochem, Beijing, China), bis[(2-diphenyl-phosphino)phenyl]ether (DPEphos; 97%, Innochem, Beijing InnoChem Science & Technology Co., Ltd., Beijing, China), 9,9-dimethyl-4,5-bis(diphenylphosphino)xanthene (xantphos; 97%, Innochem), 2,9-dimethyl-4,7-diphenyl-1,10-phenanthroline (bathocuproine, bcp; 97%, Innochem), **CoPc** (97%, *β*-form, Aldrich, Sigma Aldrich (Shanghai) Trading Co., Ltd., Shanghai, Shanghai, China), **CoTAPc** (95%, Jilin Chinese Academy of Sciences-Yanshen Technology Co., Ltd., Jilin, China), CH_3_CN (Superdry, >99.5%, Innochem), NMP (Superdry, >99.5%, Innochem), TEA (Superdry, >99.5%, Energy Chemical Co., Ltd., Anhui, China), phenol (99.5%, Aladdin, Shanghai Aladdin Biochemical Technology Co., Ltd., Shanghai, China), 2,2,2-trifluoroethanol (TFE, 99.5%, Aladdin), CO_2_ (99.999%), N_2_ (99.999%), ^13^CO_2_ (99.9%), and other chemicals were commercially available and used without further purification.

### 3.2. Instruments

Electrochemical measurements were carried out using an electrochemical workstation (CHI 620E, Shanghai, China). All potentials were referenced against ferrocenium/ferrocene (Fc^+^/Fc) as an internal standard. Unless otherwise stated, all potentials were footnoted as vs. normal hydrogen electrode (NHE) by adding 0.64 V based on the potentials vs. Fc^+^/Fc. The irradiation experiments were carried out with a blue LED light (Zolix, MLED4, Beijing, China). Gas chromatographic analysis was conducted on an Agilent 7820A gas chromatography system equipped with a thermal conductivity detector (TCD) and a TDX-01 packed column, where the oven temperature was held constant at 60 °C, and the inlet and detector temperature were set at 80 °C and 200 °C, respectively. The isotopic labeling experiment was conducted under ^13^CO_2_ atmosphere and the gas in the headspace was analyzed by a quantitative mass spectrometer attached to an Agilent 7890A gas chromatography system. UV–vis spectra were collected on a Shimadzu UV-3600 spectrophotometer. The emission experiments were conducted on a modular fluorescent life and steady-state fluorescence spectrometer (FLSP1000, Edinburgh Instruments Ltd., Livingston, UK). The TA spectroscopy was carried out on a laser flash photolysis instrument (LP980, Edinburgh Instruments Ltd.). All experiments were operated at room temperature (24~25 °C) unless otherwise stated.

### 3.3. Synthesis of **CoTDMAPc**

**CoTAPc** (0.79 mmol, 0.5 g) and iodomethane (3 mL) were stirred at room temperature in DMF (20 mL) for 24 h. After the reaction, diethyl ether was added to the mixture. The precipitate was collected by centrifugation with ethanol for five times and with water for five times, respectively. The final product was dried in an oven at 373 K as a blue-green solid. MALDI-TOF: **CoTDMAPc** (measured: 743.249; simulated: 743.251).

## 4. Conclusions

To sum up, our systematic studies on the electronic effects of β-substituents within **CoPc**-based catalysts in photocatalytic CO_2_ reduction have not only created high-performance noble-metal-free systems displaying an excellent apparent quantum yield of 63.5%, but have also revealed the volcano-like structure–activity relationship along with the increasing electron-withdrawing capacity of the β-substituents. For the mechanistic insights, firstly, as demonstrated by electrochemical results and DFT calculations, the Hirshfeld populations of the Co(II) atomic charges are positively correlated with the ability of substituents to attract electrons in the following order: -N(CH_3_)_2_ < -NH_2_ < -H < -COOH < -HSO_3_. This establishes that modifying electron-withdrawing β-substituents on **CoPc** generally leads to improved photocatalytic activity by lowering the overpotential required for CO_2_ reduction. Nonetheless, the binding energy of the catalyst to adsorb *COOH or *CO intermediates is also heavily affected by the electronic tuning; in particular, the CO-desorption step is inevitably disfavored by an excessive electron-withdrawing capacity. Given the trade-off behavior between overpotential and catalytic activity which is associated with the CO desorption reaction, the **CoPc**-based catalysts showed increasing catalytic activity in the following order: -N(CH_3_)_2_ < -NH_2_ < -H < -HSO_3_ < -COOH. Overall, we believe that the above mechanistic findings should provide valuable clues in guiding the rational design of molecular catalysts for cost-effective CO_2_ photoreduction.

## Figures and Tables

**Figure 2 molecules-29-04994-f002:**
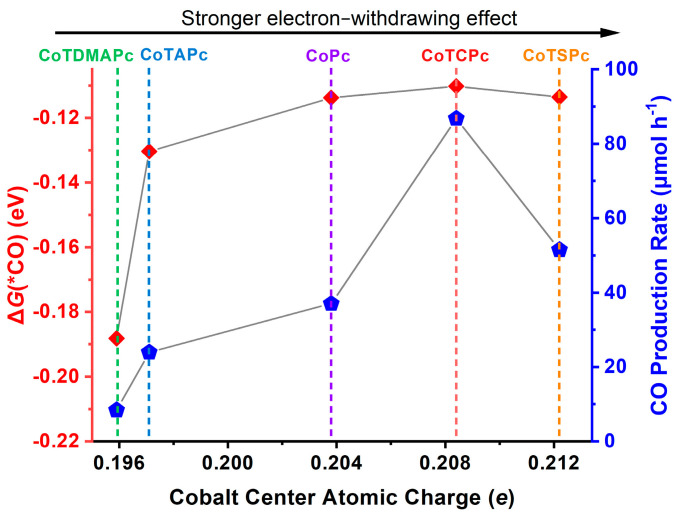
Structure-activity relationship. Plots of Δ*G*(* CO) (red diamond) and CO production rate (blue pentagon) of five Co(II) catalysts versus their Hirshfeld Co atomic charges, showing the correlations between Δ*G*(* CO) and catalytic activity as well as the volcano-like trend in electronic effects.

**Figure 3 molecules-29-04994-f003:**
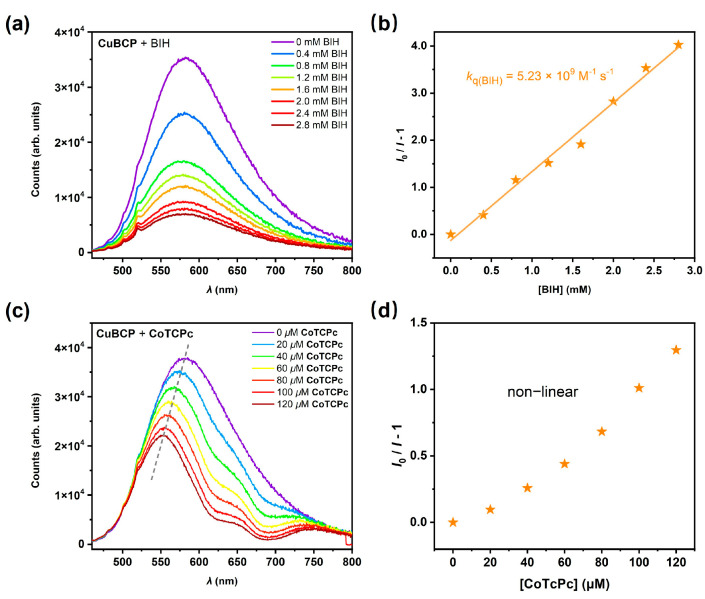
Steady-state emission quenching. Emission spectra of an Ar-purged CH_3_CN solution containing 0.2 mM **CuBCP** in the absence and presence of varying concentrations of quencher (**a**) BIH or (**b**) **CoTCPc**), respectively. Ratio of luminescence intensity versus (**c**) [BIH] with linear fitting or (**d**) [**CoTCPc**].

**Table 1 molecules-29-04994-t001:** Results of photocatalytic CO_2_ reduction experiments ^a^.

Entry	Catalyst	n(CO) (μmol)	n(H_2_) (μmol)	TON(CO)	CO%
1	**CoTDMAPc**	8.4	0.23	42	97
2	**CoTAPc**	23.9	0.56	120	98
3	**CoPc**	37.0	0.79	185	98
4	**CoTSPc**	51.5	0.95	258	98
5	**CoTCPc**	86.8	0.62	456	99
6 ^b^	**CoTCPc**	2.16	0.07	2950	97

^a^ Standard condition: **CuBCP** (0.5 mM), xantphos ligand (1.0 mM), Co(II) catalysts (0.05 mM), phenol (5.0 v%), and BIH (20 mM) in 4 mL CH_3_CN/TEA (*v*:*v* = 5:1) within 1 h of 450 nm irradiation (100 mW cm^−2^) under 1 atm CO_2_. ^b^ **CoTCPc** (0.2 μM) was used for 1 h photocatalysis and then detection.

**Table 2 molecules-29-04994-t002:** The summary of the geometric parameters of the cobalt phthalocyanine catalysts presenting the Co-N bond length (Å), the angle of plane distortion (*φ*, °), Hirshfeld Co atomic charges (*e*), and the position of 1st reduction waves (V vs. Fc^+^/Fc^0^) in their CVs.

Molecules	*d*(Co-N)	*φ*	Hirshfeld Co Atomic Charge	1st Reduction Wave
**CoTDMAPc**	1.9274	0.00	0.1959	−1.46
**CoTAPc**	1.9272	0.00	0.1971	−1.43
**CoPc**	1.9266	0.00	0.2038	−1.38
**CoTCPc**	1.9269	0.00	0.2084	−0.88
**CoTSPc**	1.9265	0.02	0.2122	−0.83

## Data Availability

The data presented in this study are available in the Appendix A.

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
