# Peer review of "Electronic Effects in Cobalt Phthalocyanine Catalysts Towards Noble-Metal-Free, Photocatalytic CO2-to-CO Reduction"

_molecules, 2024, doi:10.3390/molecules29214994_

Round 1

Reviewer 1 Report

Comments and Suggestions for Authors

The review report is attached.

Author Response

The study is interesting, and the chemistry section meets the acceptance criteria. However, there are some observations regarding the theoretical part that should be supplemented. There are missing calculations to support the proposed mechanism in Figure 4:

Step 1: where the phthalocyanine receives an electron, needs to be supported by determining Fukui indices and dual reactivity descriptors to assess the affinity and the region where the electron enters. Once the electron is received, the spin density should be determined to see where the electron is delocalized. According to the proposed mechanism, the unpaired electron should show a preference for delocalization near the cobalt atom or in the regions of the phthalocyanine.

Step 2: The interaction of CO2 with the phthalocyanine must be supported by two additional calculations. First, an Intrinsic Reaction Coordinate (IRC) must be performed to determine the activation energy and stability of the final compound. This will allow the evaluation of the kinetics and thermodynamic stability of the interaction with CO2.

Once the optimal distance between CO2 and the phthalocyanine is determined, the type of interaction between the two molecules should be assessed. For this, a Reduced Density Gradient (RDG) study should be performed, and the bond order between them should be determined. To facilitate and speed up the process for the authors, they can download the Multiwfn program, which is open-source and free (no license required). The optimized file should be loaded, and then under the analysis options, RDG should be selected. Additionally, the bond order between all atoms can be determined. The bond order between Co and C of the CO2 should be chosen.

Step 3: The Fukui indices and dual reactivity descriptors should be recalculated for the two molecules involved in the transfer of the electron and proton. Additionally, another IRC (Intrinsic Reaction Coordinate) calculation should be performed to determine the exit energies of the OH group that forms from the proton addition to CO2. The purpose is the same as before: to determine the kinetics and thermodynamic stability and assess where the equilibrium is shifted. This is important because if the equilibrium is shifted towards the formation of the species containing the OH group, the catalytic process will be much more favorable. Moreover, if its activation energy is low, the process will occur more rapidly.

Step 4: The IRC of the CO molecule's departure from the phthalocyanine should be recalculated, as well as its RDG and bond order, similar to the analysis explained in Step 2

An important aspect of electron transfer, which the authors do not discuss, is the band gap of the phthalocyanine. This should have a small band gap between 1 and 2.5 eV for efficient electron reception. A band gap above 3 eV is typical of non-conductive molecules, which makes it crucial to determine this parameter in the optimized phthalocyanine before the catalytic process begins. The band gap can be theoretically calculated as the difference between the frontier orbitals (HOMO-LUMO), or it can be experimentally determined (which is more recommended) by performing a UV-Vis spectrum of the phthalocyanine and applying the Tauc equation approximation. It can also be done by cyclic voltammetry, subtracting the onset potentials of reduction and oxidation. These techniques provide the experimental band gap with high accuracy, and the difference between both methods usually does not exceed 0.1 to 0.3 eV.

In lines 167 and 168, the authors state: "All Co atomic charges are positive, which indicates that the electron density is transferred from cobalt to the phthalocyanine substituents."

The electronic density should be determined as the HOMO² for a clearer visualization for the reader. The authors should graph the HOMO of the phthalocyanine and then square it. This will indicate where the electron density is distributed within the molecule. Additionally, this statement must be corroborated by determining the dual reactivity descriptors of the molecule in order to ascertain which regions the charge density is transferred to—whether it is indeed the substituents or other regions of the molecule.

Furthermore, the locations of the HOMO and LUMO should be determined to estimate the degree of conjugation within the molecule, which relates to the ease of charge transfer between different regions of the molecule. Without this analysis, the statement remains an unverified hypothesis.

While the proposed mechanism is reasonable, it must be validated with the
aforementioned studies; otherwise, the proposed mechanism cannot be
accepted as a theoretical study.

Reply: Thank you for your professional suggestion. The proposed photocatalytic mechanisms of CoTCPc derivatives have been confirmed by systematic calculations in our earlier study (Chinese J. Catal. 2023, 49, 160-167; JACS Au 2023, 3 (7), 1984-1997). In order to avoid needless double calculations, we only offer calculation data in this study that are sufficient to explain the structure-activity relationship of CoTCPc derivatives and remove the previous Figure 4 for a more precise description. The reviewer's suggested HOMO, LUMO, Mayer bond order, and electro density at the cobalt center were obtained with Multiwfn 3.8, whose input files were taken from the Gaussian format checkpoint. The corresponding data were provided in the Supplementary Figure 11 and Supplementary Table 3, 4 of the revised SI (Page 9 and 11). The Intrinsic Reaction Coordinate (IRC) has also been provided in Supplementary Figure 12 (Page 9 of the revised SI) to determine the activation energy and stability of the final compound.

Now we also provide the discussion of the HOMO-LUMO gap in the revised manuscript (Page 5), which is theoretically calculated as the difference between the frontier orbitals (HOMO-LUMO). The suggested method of experimentally determining the band gap using UV-vis absorption spectra, which is generally used for solid semiconductors, could be less suitable for the molecular system described in this paper.

Reviewer 2 Report

Comments and Suggestions for Authors

Comment 1: Authors should include more details about the GC conditions and the measurement of the gas sample.

Comment 2: The annotation for the x-axis in Figure 1 is missing.

Comment 3: The electronic effect of substituents decreases with the number of atoms between the reaction center and the substituent, as well as with the position of substituents on the aromatic ring. Are there specific or different effects associated with substituents in various positions on the aromatic ring?

Comment 4: Figure 2 should be presented at full scale.

Comment 5: Authors are advised to include more literature regarding the effects of substituents and the structure-activity relationship.

Author Response

Comment 1: Authors should include more details about the GC conditions and the measurement of the gas sample.

Reply: As suggested, we now add a detailed account of the gas chromatography set-up and the sampling method in the revised SI (Page S2).

Comment 2: The annotation for the x-axis in Figure 1 is missing.

Reply: Sorry for this mistake and we have revised Figure 1 in the revised SI (Page 4).

Comment 3: The electronic effect of substituents decreases with the number of atoms between the reaction center and the substituent, as well as with the position of substituents on the aromatic ring. Are there specific or different effects associated with substituents in various positions on the aromatic ring?

Reply: Substituents in the α- or β- position of the cobalt phthalocyanine aromatic ring affect its electronic effect, which is reflected in the spectrum (J. Photoch. Photobio. A, 2009, 201, 23-31), catalytic performance, stability (CCS Chemistry, 2023, 5, 1827-1840.), solubility (J. Inclusion Phenom. Macrocyclic Chem., 2015, 82, 195-202.), and so on.

This study focused solely on the electronic effects of β-substituents, as α-substituents may have additional steric effects that could reduce catalyst stability and complicate structure-activity relationships, which could beyond the scope of our studies.

Comment 4: Figure 2 should be presented at full scale.

Reply: As suggested, we have modified Figure 2 to display at full scale in the revised SI (Page 6).

Comment 5: Authors are advised to include more literature regarding the effects of substituents and the structure-activity relationship.

Reply: As suggested, we now cite more literature regarding the effects of substituents and the structure-activity relationship as Ref. 22, 23.

Round 2

Reviewer 1 Report

Comments and Suggestions for Authors

No